# Effects of Starch Molecular Structure and Physicochemical Properties on Eating Quality of *Indica* Rice with Similar Apparent Amylose and Protein Contents

**DOI:** 10.3390/foods12193535

**Published:** 2023-09-22

**Authors:** Dawei Zhu, Xin Zheng, Jing Yu, Mingxue Chen, Min Li, Yafang Shao

**Affiliations:** 1Rice Product Quality Supervision and Inspection Center, Ministry of Agriculture and Rural Affairs, China National Rice Research Institute, Hangzhou 310006, China; zhudawei@caas.cn (D.Z.); zhengxin_zxin@163.com (X.Z.); yjingyx@163.com (J.Y.); cmingxue@163.com (M.C.); 2Rice Research Institute of Guizhou Province, Guiyang 550000, China

**Keywords:** *indica* rice, eating quality, starch structure, sensory evaluation, protein composition

## Abstract

It is important to clarify the effects of starch fine structure and protein components on the eating quality of *indica* rice. In this study, seven *indica* rice varieties with similar apparent amylose content (AAC) and protein content (PC) but different sensory taste values were selected and compared systematically. It was found that except for AAC and PC, these varieties showed significant differences in starch molecular structure and protein components. Compared with rice varieties with a low sensory taste value, varieties with a higher sensory taste value showed significantly lower amylose and higher amylopectin short chains (degree of polymerization 6–12) content; the protein component showed that the varieties with good taste value had higher albumin and lower globulin and glutelin content (*p* < 0.05). Rice varieties with lower AC, globulin, and glutelin content, as well as a higher content of albumin and amylopectin short chains, resulted in a higher swelling factor, peak viscosity, breakdown value, and ratio of hardness to stickiness, in which condition cooked rice showed a higher sensory taste value. Moreover, this study indicated that rice varieties with a higher content of albumin and amylopectin short chains were conducive to the good appearance of cooked rice. This study lays the foundation for the taste evaluation of good-tasting *indica* rice.

## 1. Introduction

Improving rice quality is always an important goal for rice breeding. Rice quality includes appearance, processing, nutritional, and eating quality. Among them, eating quality is a complex trait, which is the most concerning trait among consumers. Eating quality, including the smell, appearance, texture, and taste of cooked rice, can be affected by apparent amylose content (AAC) and protein content (PC) [1,2]. Starch and protein are the most important components in rice grains, accounting for more than 80% and 6–9% of milled rice, respectively. Starch is composed of two major components: amylose and amylopectin. Previous studies suggested that rice with higher AAC is usually associated with a harder texture of cooked rice, and lower AAC can increase the viscosity of cooked rice [3]. Protein is the second main macromolecule in milled rice. Proteins affect the texture of cooked rice through a network linked by disulfide bonds that restrict the absorbance of water during the early stages of cooking [4,5]. Therefore, rice with high protein content usually has a harder texture. Okadome et al. [6] considered that the hardness of cooked rice is affected by starch, while the surface hardness is mainly related to protein content. Hence, reducing AAC and PC is an effective approach to improving the eating quality of *indica* rice. Some previous studies found that, in recent years, the AAC and PC of most *indica* rice varieties bred in southern China were stable, at 17–18% and 7–8%, respectively [7]. Recently, it was found that some *indica* rice varieties had similar AAC and PC, but they had significantly different eating qualities [8]. Therefore, it is difficult to distinguish rice varieties with different eating qualities when mainly relying on AAC and PC. 

In addition to AAC and PC, amylopectin chain-length distribution can also affect rice quality [9,10]. For example, it has been found that medium-length amylopectin chains (DP 15–25) positively correlate with pasting temperature, while longer amylopectin chains (DP > 36) positively correlate with the final viscosity [9,11]. It was indicated that long amylopectin chains were positively linked with cooked rice hardness, while short amylopectin chains (DP < 70) showed significant negative correlations with hardness, which means rice varieties with high and long amylopectin chains had harder textures [12]. Li et al. [12] proved that a high proportion of long amylose chains (DP 1000–2000) was an important factor that could cause the harder texture of cooked rice. Moreover, other starch fine structures, such as starch granule size, were reported to be important parameters affecting the appearance of cooked rice [13]. Rice proteins can be classified into albumin, globulin, prolamin, and glutelin according to their solubility [14]. Previous studies demonstrated that prolamin and glutelin affected the texture of cooked rice significantly [14,15]. Furthermore, a Rapid Visco Analyzer (RVA) and texture analyzer have often been used to study rice eating quality. As reported, the RVA profile, including elements such as peak viscosity and final viscosity, negatively correlated with the hardness of cooked rice, and rice with a lower setback value was more resistant to retrogradation after cooling [16]. The characteristics of cooked rice texture (e.g., hardness, stickiness, and springing) could well reflect the eating quality of cooked rice [12].

In order to better understand the effects of starch molecular structure and physicochemical properties on the eating quality of *indica* rice, seven *indica* rice varieties with similar AAC and PC were selected, and starch molecular structure, physicochemical properties, rice eating quality, and milled-rice grain shape were analyzed. This is because previous studies mainly focused on the effects of physicochemical properties on rice eating quality while ignoring the influence of milled-rice grain shape. Few studies have reported the effects of milled rice appearance on rice eating quality, yet rice is cooked in whole grains instead of as rice flour. It would help to understand the formation mechanism behind the eating quality of rice varieties with similar apparent amylose and protein contents and provide new information for the breeding of high-quality *indica* rice varieties.

## 2. Materials and Methods

### 2.1. Materials

Seven *indica* rice varieties (Nongxiang 42 (NX 42), Yexiangyouxinhuazhan (YXYXHZ), Yueliangyou 2646 (YLY 2646), Diangu 163 (DG 163), Jingyouzhan (JYZ), Xinyou399 (XY 399), and Quanyousimiao (QYSM)) were used. All rice varieties were planted in 2021. The matured paddy rice was harvested and air-dried after removing impurities. The moisture content of paddy rice was controlled at 13–14%.

### 2.2. Milled Rice Appearance Characteristics

About 35 g of the milled rice sample was placed on the glass plate of a scanner (EPSON V600, Nagano-ken, Japan). We ensured that the complete milled rice grains were spread out without overlapping, and then they were scanned to obtain an image of the sample. The results of grain length, length/width ratio, chalkiness degree, and transparency were obtained by using rice-appearance quality analysis software (WANSHEN SC-E, Hangzhou, China).

### 2.3. Rice Flour and Starch Extraction

The paddy rice was shelled with a rice husker (SY88-TH, Gangwon-do, Korea) and then milled by a grain polisher (Kett, Wuxi, China). Rice flour was prepared using a mill (FOSS 1093, Aalborg, Denmark). Starch was extracted from rice flour according to the method reported by Wei et al. [17]. Briefly, about 10 g of milled rice flour was treated with alkaline protease (Solarbio, Shanghai, China) to remove the protein. The homogenate was collected with a 200-mesh sieve, and then 30 mL of deionized water was added. The upper impurities were removed after centrifuging at 4000× *g* for 10 min. This was repeated five times to ensure that the impurities were removed completely.

### 2.4. Determination of the Apparent Amylose Content and Protein Content

The apparent amylose content (AAC) was determined according to the method reported by Man et al. [18]. About 10 mg of a dried starch sample was weighed into a centrifuge tube, urea-dimethyl sulfoxide solution (UDMSO) was added, and then it was dissolved in a water bath at 95 °C for 1 h. We placed 1 mL of starch-UDMSO solution into a 50 mL volumetric flask, added water to nearly 50 mL, and followed by adding 1 mL of 0.2% I2, 2% KI (*w/v*) staining solution. Then, we fixed the volume, mixed the solution well, and placed it in the dark for 10 min. The absorbance of the sample was measured at 620 nm, and then the standard curve was prepared using the standard solutions of potato amylose and maize amylopectin. Protein content (PC) was determined according to the American Association of Cereal Chemistry (AACC) 1984 procedures [19]. 

### 2.5. Taste Sensory Evaluation

Taste sensory evaluation was performed according to the Chinese National Standard (GB/T 15682-2008 “Inspection of grain and oils–Method for sensory evaluation of paddy or rice cooking and eating quality”). The taste sensory evaluation team consisted of seven people of different genders and ages who were trained professionally to identify rice eating quality. The reference sample was Yuzhenxiang (*indica* rice, 90 scores). Taste sensory evaluation included five aspects: cooked rice fragrance, appearance, palatability, taste, and cold rice texture. Each aspect included seven options: 0 = as control, 1 = slightly good, 2 = better, 3 = best, −1 = slightly poor, −2 = worse, −3 = worst. The taste sensory evaluation was in accordance with the Declaration of Helsinki.

### 2.6. Protein Composition (Albumin, Globulin, Prolamin, and Glutelin) Analysis

According to the method used by Baxter et al. [14], first the milled rice (about 5 g) was defatted with 20 mL hexane and dried under a hood at 25 °C for one day. Then, albumin, globulin, prolamin, and glutelin were sequentially extracted by deionized water, 2% sodium chloride solution, 60% ethanol solution, and 0.0025 mol/L NaOH solution, respectively. All the extractions were repeated three times, and the protein content was measured using bovine serum albumin as standard using a protein assay kit (Solarbio, Shanghai, China). 

### 2.7. Starch Molecular Size Distribution

Pure starch was first debranched with isoamylase (Megazyme E-ISAMY, Wicklow, Ireland) and then freeze-dried according to the method reported by Wu et al. [20]. The pure starch (4 mg) was dispersed with 0.9 mL deionized water and then mixed with isoamylase (2.5 μL, 1000 U/mL), acetate buffer solution (0.1 mL, 0.1 M, pH 3.5) and sodium azide solution (5 μL, 0.04 g mL^−1^). The mixture was incubated at 35 °C for 3 h. The debranched starch was first neutralized with a 0.1 M NaOH solution and then heated in a water bath for 2 h and freeze-dried overnight for SEC analysis. The molecular weight distribution of debranched starch was determined using an LC-20 CE Shimadzu system equipped with an RID-10A refractive index detector (Shimadzu Corporation, Kyoto, Japan), as Gu et al. [21] described. The flow rates were controlled at 0.6 mL/min, and a combination of GRAM100 and 1000 analytical columns (PSS, Mainz, Germany) was used. 

### 2.8. Amylopectin Fine Structure Analysis

Starch was first debranched and freeze-dried using the above method described by Wu et al. [20]. The amylopectin fine structure was measured with a PA-800 Plus FACE System (Beckman-Coulter, Brea, CA, USA) equipped with a solid-state laser-induced fluorescence (LIF) detector [21]. Testing conditions were set as follows: The injection volume was 50 μL, the column temperature was 25 °C, and the flow rate was 0.5 mL/min. The progressive process for the mobile phase was: 40% B in 0 min; 50% B in 2 min; 60% B in 10 min; 80% B in 40 min; and maintained for 20 min (mobile phase A: 150 mmol/L NaOH; mobile phase B: 150 mmol/L NaOH with 500 mmol/L sodium acetate).

### 2.9. Swelling Power and Water Solubility

The swelling power and water solubility were measured according to the method reported by Konik-Rose et al. [22] with minor modifications. About 30–40 mg of sample (m0) was weighed and put into a 2 mL centrifuge tube (m1) that had been weighed, combined with 1 mL of ultrapure water, shaken in a water bath at 90 °C for 1 h, and centrifuged at 4000×g for 10 min. Then, its total weight (m2) was measured after drawing the supernatant with a pipette gun, and this was followed by drying at 60 °C and weighing again (m3). Solubility (%) = 100 × (m0 + m1 − m3)/m0 × 100%; Swelling power (g/g) = (m2 − m1)/(m3 − m1).

### 2.10. Pasting Properties

Three grams of rice flour sample was weighed in an aluminum can and added to 25 mL of distilled water. The test procedure was performed according to the American Association of Cereal Chemistry (AACC) 1984 procedures [19]. First, the temperature was maintained at 50 °C for 1 min, then heated to 95 °C and maintained at 95 °C for 5 min, followed by cooling to 50 °C and maintained at 50 °C for 2 min. During this, both the heating and cooling rates were 6 °C/min, and the stirring rate was 960 r/min for the first 10 s, then rapidly decreased to 160 r/min and kept constant. The RVA profile characteristics were expressed in cP in terms of peak viscosity, through viscosity, final viscosity and breakdown viscosity, and setback viscosity.

### 2.11. Textural Properties of Cooked Rice

The textural properties of cooked rice were measured using a TA.XT–Plus texture analyzer (Stable Micro Symain Stems Ltd., Surrey, UK) according to the method reported by Li et al. [12]. About 30 g of milled rice was cooked under the same conditions, and approximately 3–5 g of cooked rice was used to measure the textural properties. The probe was P/36R with a sensing force of 5 g; pre-test speed: 10 mm/s; test speed: 0.5 mm/s; post-test speed: 5 mm/s; compression ratio: 50%; time interval between two compressions: 5 s. Each sample was measured six times. 

### 2.12. Statistical Analysis

One-way analysis of variance and Tukey’s post-hoc tests were conducted using SPSS 16.0 statistical software. Principal component analysis (PCA) of the starch fine structure, protein components, and physicochemical properties was performed using Origin 2021. We determined the Pearson correlation coefficient and carried out graphing using Origin 2021.

## 3. Result and Discussion 

### 3.1. AAC, PC and Sensory Taste Value 

AAC and PC showed no significant difference among seven *indica* rice varieties, while significant differences were observed in sensory taste value (Figure 1). In this study, rice varieties with a taste value greater than or equal to 85 marks were recognized as a good-taste-value varieties group (GTVG), and those below 85 marks were marked as a medium- or poor-taste-value varieties group (MPTVG). Thus, NX42, YXYXHZ, and XY399 belonged to the GTVG, while QYSM, YLY2646, DG163, and JYZ belonged to the MPTVG. The average sensory taste value of the GTVG was 12.2% higher than that of the MPTVG. This was attributed to the higher fragrance, appearance, palatability, taste, texture of cooked rice, and texture of cooled cooked rice (Table 1). The AAC and PC are usually recognized as important factors that could affect rice eating quality. It was reported that a higher AAC was negatively correlated with taste value [23], and rice with a higher AAC caused a harder texture compared with those with a lower AAC [24]. Protein could influence the cooked rice texture due to a network linked by disulfide bonds [25]. Zhu et al. [13] used *japonica* rice varieties with similar AACs as materials and found that a higher PC could decrease the eating quality. In recent years, with the development of high-quality rice breeding in China, the amylose content of *indica* rice varieties has decreased and stabilized at about 17–18% [7]. The existence of *indica* rice varieties with a similar AAC but different sensory taste values might be caused by their different true amylose content, as AAC was the content of amylose and several long-branch amylopectin chains [26]. Although the average PC of the MPTVG was slightly higher than that of the GTVG, the difference was insignificant. These results indicated that it was inadequate to evaluate the eating quality of *indica* rice varieties using these two parameters.

### 3.2. Appearance Quality of Milled Rice

Grain length (GL), length/width ratio (LWR), chalkiness degree, and transparency ranged from 6.5 to 7.9 mm, 3.2 to 4.3, 0.1 to 2.7%, and grade 1 to 2, respectively (Figure 2). The average GL and LWR of the GTVG were 6.5% and 29.3% higher than those of the MPTVG, respectively. The average chalkiness degree and transparency of the GTVG were 74.5% and 33.3% lower than those of the MPTVG, respectively. It is suggested that the milled rice of the GTVG had a slender shape and a more transparent appearance. A recent study suggested that *indica* rice with a higher LWR was beneficial for obtaining higher eating quality [7]. Similar results have also been found in *japonica* rice [27]. This might be because the slim rice grains with similar AAC and PC could be heated more evenly during cooking, as the shape of the milled rice grains was an important factor affecting water absorption and swelling [28].

### 3.3. Rice Protein Components

Although all rice samples had similar PC, their protein components varied significantly (Figure 3). Among the four protein components, glutelin was the most abundant (20.9–26.6 mg/g), and prolamin was the least (0.11–0.70 mg/g). It was found that the GTVG rice varieties had higher albumin content and lower globulin, prolamin and glutelin contents compared with the MPTVG (*p* < 0.05). Liang et al. [29] suggested that increased contents of globulin and glutenin in rice grains could be caused by applying nitrogen fertilizer, which would lead to an increase in the hardness of cooked rice. This is mainly because globulin and glutenin affected the texture of the gel and cooked rice by interacting with rice starch [4]. Therefore, rice varieties with higher globulin and glutenin contents would decrease the palatability of cooked rice and eventually decrease the sensory taste value. It was also reported that water-soluble albumin could influence the texture of cooked rice by covering the surface of cooked rice during cooking [30], which disagreed with our study. This was probably due to the fact that albumin was shown to increase the surface hardness of cooked rice, but the direct impact of albumin on the mouthfeel might be smaller because it was present in smaller amounts compared to glutenin in rice. Moreover, a previous study also proposed that higher water-soluble albumin content in rice grains was an important factor in improving the appearance of cooked rice [31], which was in accordance with our results on the appearance of cooked rice in the GTVG. In this study, we found that *indica* rice varieties with a higher sensory taste value had a relatively lower glutelin content. However, this does not mean that the lower glutelin content in rice grains could cause higher eating quality. Liu et al. [32] studied 105 *japonica* rice varieties with AAC ranging from 13% to 20% and found that PC was significantly negatively correlated with taste value, which was in accordance with our findings. However, PC showed no significant correlation with taste value when AAC was lower than 13%. In addition to rice eating quality, PC in rice grains also affected nutritional value [33]. In this study, the conclusion about glutelin content only refers to eating quality.

### 3.4. Starch Molecular Structure

The molecular weights of total starch ranged from 1 to ~10,000 DP, with three peaks (Figure 4) representing short- and long-branch chains of amylopectin and amylose, respectively. The amylose could be classified into short- and long-amylose-chain fractions according to a previous study [34]. Therefore, amylose content (AC) was determined as the ratio of the area under the curve of the amylose branches to that of the entire distribution. The AC, ratio of amylose to amylopectin (AC/AP), the ratio of short amylose chains to long amylose chains (AM1/AM2), and the ratio of short amylopectin chains to long amylopectin chains (AP1/AP2) of these varieties ranged from 13.7 to 18.0%, 0.16 to 0.22, 1.99 to 3.74 and 3.71 to 4.36, respectively (Table 2). The AC (determined by SEC) was 2.0–24.0% lower than AAC, which was mainly because some long-branch amylopectin chains could also bind with iodine in the measurement of AAC. It was found that the AC and AM/AP of the GTVG varieties were lower than those of the MPTVG. This was because AC was an important factor affecting rice eating quality. Rice varieties with high AC usually had a poor taste value, which might be because amylose could inhibit the swelling of starch granules and maintain the integrity of swollen granules during heating [35,36]. Similar results were also found by Peng et al. [8], who used *indica* hybrid rice and parental lines with similar AAC but different taste values as materials and found that AC was an important factor affecting the taste value of cooked rice. Except for AC and AM/AP, significant differences between AP1/AP2 and AM1/AM2 among the seven *indica* rice varieties were observed, with no obvious pattern between the GTVG and the MPTVG. The results of starch fine structures suggested that rice varieties with similar AAC may have significantly different starch fine structures and that the variation in rice eating quality was mainly caused by AC.

### 3.5. Amylopectin Chain-Length Distributions

In order to better analyze the differences in amylopectin fine structure among varieties, amylopectin chain-length distributions were measured by fluorophore-assisted capillary electrophoresis (FACE). It showed that the branched-chain ranged from 6 to 100 DP. The amylopectin chain-length distributions were classified into four fractions (Fa, Fb_1_, Fb_2_, and Fb_3_), as in a previous study [37]. The content of Fa, Fb_1_, Fb_2_ and Fb_3_ varied from 27.1% to 29.4%, 47.9% to 49.7%, 9.5% to 10.3%, and 12.8% to 13.8%, respectively (Table 3). Amylopectin chain-length distributions were significantly different among rice varieties. The Fa content of the GTVG was significantly higher than that of the MPTVG (*p* < 0.05). This was mainly because a higher proportion of Fb_1_ or AP1 could increase the stickiness of cooked rice and improve its mouthfeel [12]. Zhu et al. [31] compared two types of *japonica* rice, high AAC with low PC and low AAC with high PC, and concluded that high and short amylopectin content could create a looser gel network and softer-cooked rice. Peng et al. [8] chose hybrid *indica* rice with similar AAC as a material and found that rice with good taste values had a high proportion of intermediate chains (DP 13–24). Ayabe et al. [38] and Tao et al. [36] considered that the fraction of short AP chains (DP 6–11) or AP1 was positively correlated with the taste value. Except for the mouthfeel of cooked rice, the existence of a higher proportion of Fa could also enhance the eating quality by improving the cooked rice’s appearance. This was because Fb_1_, the main component of the dissolved solids, could be dissolved from the rice grains during cooking [39], and it covered the surface of the rice grains to form a thin film. Therefore, it was Fb_1_ that gave the cooked rice a shiny appearance. 

### 3.6. Swelling Factors of Rice Starch 

The swelling factors include swelling power and water solubility, which reflected the water-holding capacity of starch molecules by hydrogen bonding during heating. The swelling power and water solubility ranged from 9.3% to 13.3% and 8.6% to 14.8%, respectively (Figure 5A,B). The starch swelling factors of the GTVG were significantly higher than that of the MPTVG. The swelling factors of pure starch were mainly determined by starch fine structures including AM and AP. Previous studies suggested that both amylose and long-chain amylopectin could inhibit starch granule swelling by maintaining the integrity of swollen granules [35]. In this study, XY 399 had a lower Fb_3_ content compared to JYZ, with no significant difference, but the AC of XY 399 was 24.1% lower than that of JYZ. Therefore, it was concluded that higher swelling factors were mainly caused by lower AC. 

### 3.7. Pasting Properties of Rice Flour

Pasting properties showed significant differences among all the samples. Peak viscosity (PV) was the maximum viscosity of rice flour during heating in water, which ranged from 2240 to 2741 cP (Table 4). The PV of the GTVG was significantly higher than that of the MPTVG. A previous study suggested that high PV was mainly attributed to low AAC [40], while in this study, the AAC of seven *indica* rice varieties showed no significant difference. Hence, the higher PV of the GTVG was mainly caused by lower AC and higher Fa content. As mentioned above, amylose could inhibit starch granule swelling by maintaining the integrity of swollen granules, and short amylopectin chains would allow starch granule to swell more easily due to weaker binding forces [35,41]. Unlike pure starch, proteins and lipids and the interactions between them should be considered, because they could affect the pasting properties of rice flour. Compared with the GTVG, the MPTVG had a higher average glutelin and globulin contents. Higher glutelin and globulin contents could compete with more water molecules, which reduced the amount of water available for starch hydration and ultimately result in lower peak viscosity [4]. BV and SV were important indices affecting eating quality. The breakdown value (BV) was used to measure the disruption of the swelling starch structure, while the setback value (SV) was used to measure how readily cooked rice underwent retrogradation upon cooling. Generally, high BV and low SV indicated that the cooked rice had a good eating quality with a relatively soft texture [42]. In accordance with previous studies, the average BV and SV of the GTVG were, respectively, 18.0% higher and 44.0% lower than those of the MPTVG, respectively. A higher BV meant that these swollen granules could not be maintained during heating, mainly due to lower AC and glutelin content [43,44]. 

### 3.8. Textural Properties of Cooked Rice

Hardness and stickiness were the most important textural properties of cooked rice. Rice varieties with a high eating quality generally had moderate hardness and high stickiness [12]. The hardness, stickiness, and hardness/stickiness ratio of cooked rice ranged from 204 to 270 g, 450 to 860 g, and 0.24 to 0.56, respectively (Figure 5C–E). As compared to the MPTVG, the GTVG had a significantly lower hardness of cooked rice and hardness/stickiness ratio and a significantly higher stickiness of cooked rice. These results were consistent with a previous study that found hybrid rice varieties with high taste values exhibited relatively high stickiness and low hardness [8]. It was suggested that AAC was the main factor affecting rice texture, as AAC was positively correlated with hardness and negatively correlated with the stickiness of rice [45,46]. In this study, the seven *indica* varieties had similar AAC and PC, and their rice/water ratio during cooking was the same. It was reported that the rice/water ratio was also an important factor influencing the texture of cooked rice [12]. Therefore, the cooked rice texture was affected by starch fine structure and protein composition. During cooking, rice expansion caused starch granule ruptures, which made the cooked rice less hard. Meanwhile, amylose, some AP long chains, and glutelin could interact with each other and restrict starch swelling [47]. Li et al. [12] used rice with different ACs (from 1.4% to 29.5%) as materials and suggested that AC was one of the most important parameters affecting the hardness of cooked rice. Therefore, the lower cooked rice hardness of the GTVG was mainly attributed to lower AC. Rice stickiness was caused by leached AM and AP chains, which were mainly affected by AP content [48]. Previous studies suggested that the high amount of short amylopectin chains would create a greater opportunity for bonding and molecular interactions in which case greater forces were required to separate the rice grains, eventually leading to higher stickiness [49]. Hence, the higher cooked rice stickiness of the GTVG was mainly caused by higher amylopectin Fa content.

### 3.9. Principal Component Analysis (PCA) of the Physicochemical Properties of Rice Varieties

PCA plots were used to visualize the differences and similarities between the good-taste-value group (GTVG) and medium- or poor-taste-value group (MPTVG). The first two principal components accounted for 65% of the total variance, with the first principal component (PC1) and the second principal component (PC2) accounting for 48.3% and 17.1% of the total variance, respectively (Figure 6). The MPTVG showed a wider variance compared with the GTVG. PC1 was mainly linked to sensory taste value, AC, glutenin content, and ratio of hardness to stickiness. PC2 was mainly linked to Fa, Fb_2_ contents, pasting temperature, and chalkiness degree. According to factor loading and sample score maps, it was suggested that sensory evaluation characteristics were the most effective parameters to distinguish the eating quality of *indica* rice varieties with similar AAC and PC. Furthermore, starch fine structure (AC, Fa, etc.), glutenin content, and the ratio of hardness to stickiness could also be used to assess the eating quality of *indica* rice varieties.

TT, taste; STV, sensory taste value; Glu, glutelin; CD, chalkiness degree; SP, swelling power; PT, pasting temperature; Har/Sti, ratio of hardness to stickiness.

### 3.10. Correlation Analysis between Milled Rice Appearance, Chemical Composition, Physicochemical Properties, and Rice Eating Quality

Figure 7 shows the correlation analysis between physicochemical properties and cooked rice sensory evaluation characteristics. Sensory taste value was positively (r = 0.905, 0.839, 0.804, 0.875, *p* < 0.05) correlated with starch swelling factors, peak viscosity, and cooked rice stickiness, and negatively (r = −0.836, −0.833, −0.829, −0.880, *p* < 0.05) correlated with AC, AM/AP, cooked rice hardness, and ratio of hardness to stickiness. This is in agreement with a previous study using *japonica* rice varieties, which found that peak viscosity was positively associated with sensory value [13]. Consistent findings were also reported by Li et al. [12], who found that amylose was significantly positively associated with cooked rice hardness and that higher amylose content in rice grains would reduce the rice eating quality. For the starch fine structure, we found that AC and AM/AP was negatively correlated with swelling factors, peak viscosity, and cooked rice stickiness, and positively correlated with pasting temperature and cooked rice hardness. Moreover, short amylopectin content showed a positive correlation with stickiness, which was proved by previous studies using both *indica* and *japonica* rice varieties [12,13]. The amylopectin showed a significant negative (r = −0.788, *p* < 0.05) correlation with stickiness, which means the existence of long-chain amylopectin would decrease the stickiness. 

## 4. Conclusions

In this study, we analyzed the starch molecular structure and physicochemical properties of seven *indica* rice with similar apparent amylose and protein content but different eating qualities. The results of the starch molecular structure analysis indicated that the AC of high-taste-value *indica* rice was low and amylopectin short-chain content was high, which led to higher swelling factors, peak viscosity, breakdown value, and cooked rice stickiness, and a lower setback value. For the protein component, the results indicated that *indica* rice with a high taste value had low globulin and glutelin content and high albumin content, which would be beneficial for reducing cooked rice hardness. These indicators could be used to distinguish the eating quality of *indica* rice varieties with similar AAC and PC. We analyzed the effects of starch molecular structure and protein components on rice eating quality for the first time, which provides new insights into the identification of the eating quality of *indica* rice.

## Figures and Tables

**Figure 1 foods-12-03535-f001:**
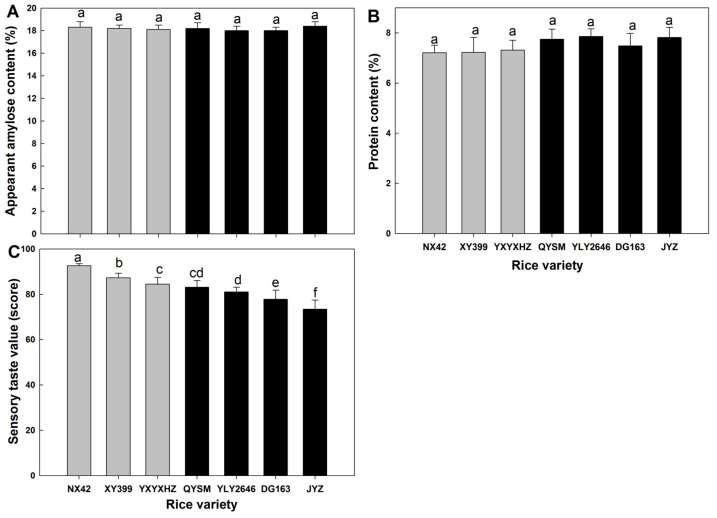
Apparent amylose content (**A**), protein content (**B**), and sensory taste value (**C**) of *indica* rice varieties. Values in the same image with different letters are significantly different (*p* < 0.05). Grey represents varieties belonging to the GTVG and black represents varieties belonging to the MPTVG.

**Figure 2 foods-12-03535-f002:**
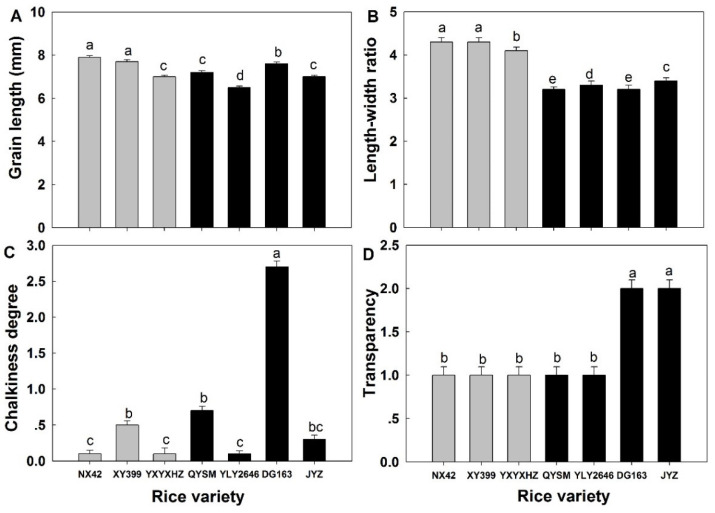
Milled rice appearance quality (**A**–**D**) of *indica* rice varieties. Values in the same image with different letters are significantly different (*p* < 0.05). Grey represents varieties belonging to the GTVG and black represents varieties belonging to the MPTVG.

**Figure 3 foods-12-03535-f003:**
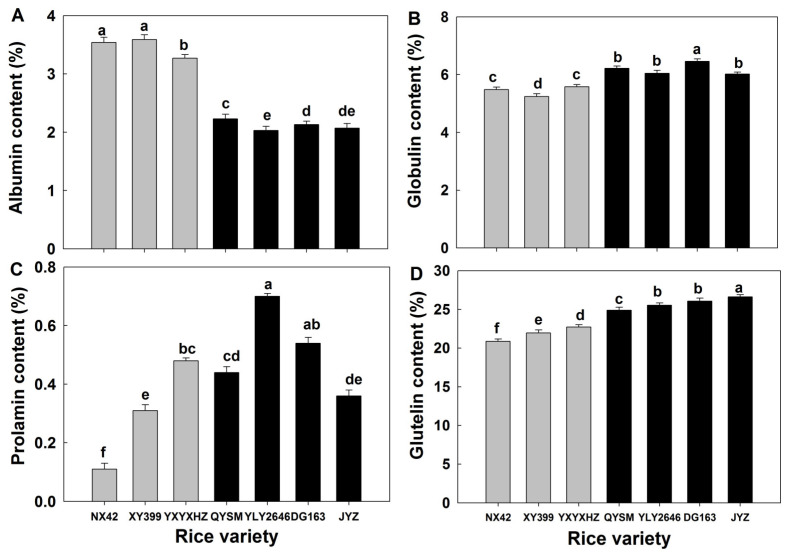
Protein component (**A**–**D**) of *indica* rice varieties. Values in the same image with different letters are significantly different (*p* < 0.05). Grey represents varieties belonging to the GTVG and black represents varieties belonging to the MPTVG.

**Figure 4 foods-12-03535-f004:**
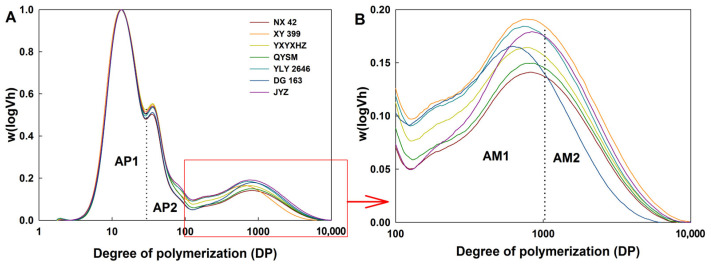
Fine structure of debranched starch from *indica* rice varieties (**A**,**B**) showed the CLD above DP 100.

**Figure 5 foods-12-03535-f005:**
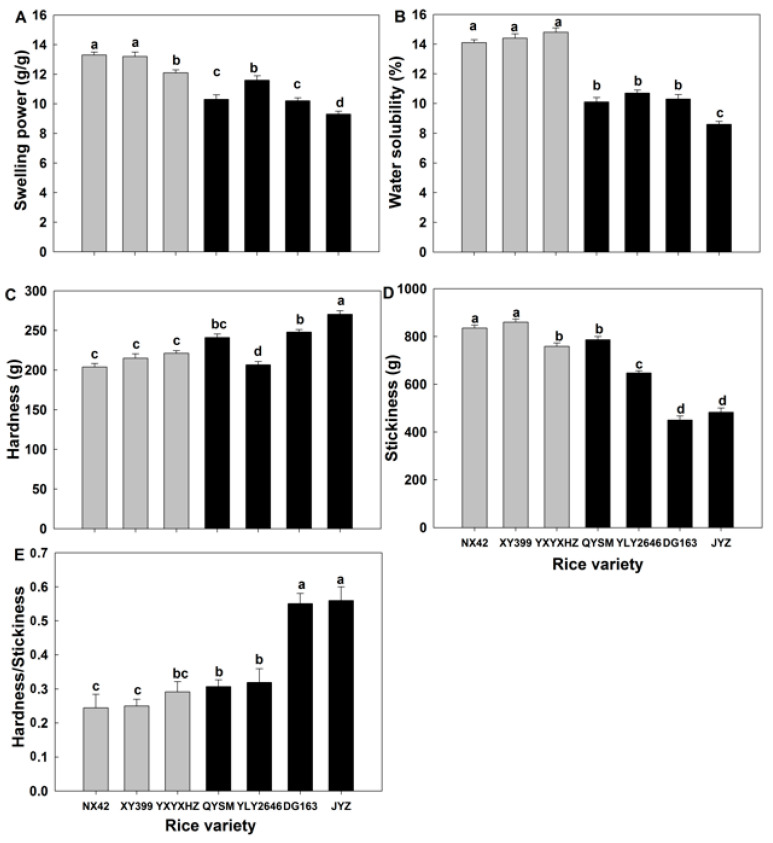
Starch swelling factors (**A**,**B**) of *indica* rice, cooked rice texture (**C**–**E**) of *indica* rice. Values in the same image with different letters are significantly different (*p* < 0.05). Grey and black represent varieties belonging to the GTVG and MPTVG, respectively.

**Figure 6 foods-12-03535-f006:**
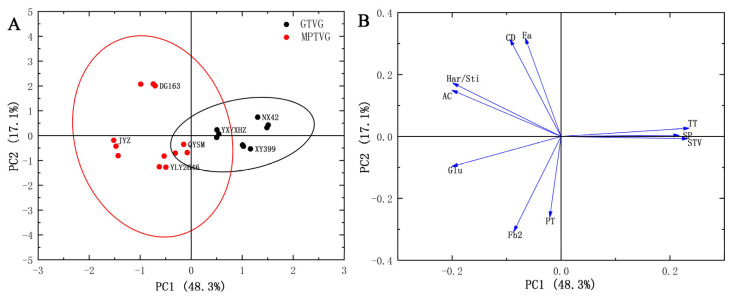
The PCA score plot (**A**) and loading plot (**B**) of *indica* rice varieties.

**Figure 7 foods-12-03535-f007:**
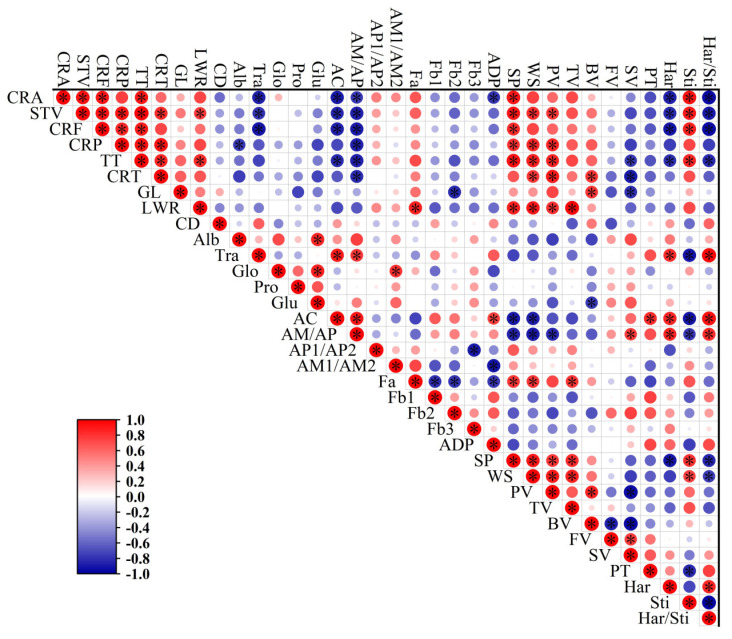
Pearson correlation of chemical composition, starch fine structure, physicochemical properties, and cooked rice sensory evaluation characteristics of *indica* rice varieties. STV, sensory taste value; CRF, cooked rice fragrance; CRA, cooked rice appearance; CRP, cooked rice palatability; TT, taste; CRT, cold rice texture; GL, grain length; LWR, length/width ratio; CD, chalkiness degree; Tra, transparency; Alb, albumin; Glo, globulin; Pro, prolamin; Glu, glutelin; ADP, amylopectin average degree of polymerization; Har, hardness; Sti, stickiness; Har/Sti, hardness/stickiness. The red and blue circles indicate negative or positive correlations between parameters, respectively; the darker the color is, the higher the correlation is. * indicates the significant differences at *p* < 0.05 levels. r_0.05_ = 0.7545.

**Table 1 foods-12-03535-t001:** Sensory evaluation characteristics of *indica* rice varieties.

Type	Variety	Sensory Taste Value	Cooked Rice Fragrance	Cooked Rice Appearance	Cooked Rice Palatability	Taste	Cold Rice Texture
GTVG	NX42	93 ± 3.4 ^a^	90 ± 4.2 ^a^	94 ± 2.7 ^a^	96 ± 4.4 ^a^	96 ± 3.6 ^a^	88 ± 4.2 ^a^
	XY399	87 ± 5.7 ^b^	86 ± 3.7 ^ab^	92 ± 4.4 ^a^	84 ± 4.1 ^bc^	90 ± 3.7 ^b^	82 ± 3.6 ^a^
	YXYXHZ	86 ± 5.5 ^c^	86 ± 3.6 ^bc^	84 ± 2.9 ^b^	90 ± 4.0 ^b^	86 ± 4.4 ^b^	86 ± 3.8 ^a^
MPTVG	QYSM	83 ± 4.3 ^cd^	86 ± 2.8 ^bc^	84 ± 3.4 ^b^	82 ± 3.8 ^cd^	82 ± 4.3 ^c^	82 ± 3.4 ^a^
	YLY2646	81 ± 4.6 ^d^	86 ± 3.1 ^bc^	86 ± 3.2 ^b^	80 ± 3.6 ^cd^	80 ± 4.1 ^cd^	74 ± 4.0 ^ab^
	DG163	78 ± 4.4 ^e^	80 ± 3.8 ^cd^	74 ± 3.4 ^c^	80 ± 3.9 ^d^	78 ± 3.7 ^d^	80 ± 4.1 ^ab^
	JYZ	73 ± 4.7 ^f^	76 ± 4.2 ^d^	74 ± 4.0 ^c^	72 ± 3.2 ^e^	72 ± 4.3 ^e^	72 ± 4.4 ^b^

Data are expressed as mean ± SD. Values in the same column with different letters are significantly different (*p* < 0.05).

**Table 2 foods-12-03535-t002:** Size exclusion chromatography (SEC) parameters of debranched starch from *indica* rice varieties.

Type	Variety	AM/(AM + AP)	AM/AP	AP1/AP2	AM1/AM2
GTVG	NX42	14.8 ± 0.12 ^cd^	0.17 ± 0.006 ^c^	4.32 ± 0.04 ^a^	2.21 ± 0.07 ^c^
	XY399	13.7 ± 0.08 ^e^	0.17 ± 0.006 ^bc^	4.26 ± 0.05 ^a^	3.74 ± 0.07 ^a^
	YXYXHZ	14.6 ± 0.10 ^d^	0.16 ± 0.006 ^c^	4.02 ± 0.03 ^b^	2.02 ± 0.04 ^d^
MPTVG	QYSM	15.5 ± 0.11 ^c^	0.18 ± 0.008 ^bc^	3.71 ± 0.07 ^c^	2.42 ± 0.05 ^b^
	YLY2646	15.3 ± 0.09 ^c^	0.18 ± 0.007 ^c^	4.36 ± 0.06 ^a^	2.52 ± 0.06 ^b^
	DG163	17.0 ± 0.09 ^b^	0.19 ± 0.008 ^b^	4.06 ± 0.06 ^b^	1.99 ± 0.06 ^d^
	JYZ	18.0 ± 0.10 ^a^	0.22 ± 0.005 ^a^	3.99 ± 0.07 ^b^	2.41 ± 0.06 ^b^

Data are expressed as mean ± SD. Values in the same column with different letters are significantly different (*p* < 0.05). AM, amylose; AM1, short amylose chains; AM2, long amylose chains; AP1, short amylopectin branch chains; AP2, long amylopectin branch chains.

**Table 3 foods-12-03535-t003:** Amylopectin chain-length distributions of starch from *indica* rice varieties.

Type	Variety	Chain-Length Proportion of Amylopectin (%)				Average Chain Length (DP)
		Fa (DP 6–12)	Fb_1_ (DP 13–24)	Fb_2_ (DP 25–36)	Fb_3_ (DP 37+)	
GTVG	NX42	28.1 ± 0.16 ^b^	49.3 ± 0.14 ^b^	9.8 ± 0.13 ^e^	12.8 ± 0.20 ^b^	21.0 ± 0.21 ^a^
	XY399	29.4 ± 0.14 ^a^	47.9 ± 0.12 ^e^	9.5 ± 0.17 ^f^	13.1 ± 0.24 ^a^	20.8 ± 0.16 ^a^
	YXYXHZ	28.0 ± 0.15 ^b^	48.5 ± 0.09 ^d^	10.2 ± 0.16 ^bc^	13.3 ± 0.19 ^a^	21.0 ± 0.19 ^a^
MPTVG	QYSM	27.2 ± 0.18 ^c^	49.0 ± 0.17 ^c^	10.1 ± 0.15 ^cd^	13.8 ± 0.18 ^a^	21.0 ± 0.18 ^a^
	YLY2646	27.1 ± 0.19 ^c^	49.6 ± 0.08 ^a^	10.2 ± 0.14 ^ab^	13.1 ± 0.22 ^a^	21.0 ± 0.12 ^a^
	DG163	27.2 ± 0.17 ^c^	49.7 ± 0.12 ^a^	9.9 ± 0.19 ^de^	13.2 ± 0.21 ^a^	21.1 ± 0.24 ^a^
	JYZ	27.2 ± 0.15 ^c^	49.2 ± 0.15 ^b^	10.3 ± 0.18 ^a^	13.3 ± 0.18 ^a^	21.1 ± 0.25 ^a^

Data are expressed as the mean ± SD. Values in the same column with different letters are significantly different (*p* < 0.05).

**Table 4 foods-12-03535-t004:** Pasting properties of rice flour from *indica* rice varieties.

Type	Variety	PV (cP)	TV (cP)	BV (cP)	FV (cP)	SV (cP)	PT (°C)
GTVG	NX42	2741 ± 5.6 ^a^	1801 ± 5.1 ^a^	940 ± 4.8 ^b^	3136 ± 4.6 ^bc^	395 ± 2.4 ^f^	88.4 ± 1.2 ^a^
	XY399	2693 ± 4.2 ^b^	1813 ± 5.1 ^a^	880 ± 4.2 ^b^	3108 ± 4.0 ^cd^	415 ± 2.3 ^f^	87.9 ± 1.1 ^b^
	YXYXHZ	2673 ± 3.7 ^b^	1814 ± 4.8 ^a^	859 ± 5.2 ^b^	3180 ± 4.5 ^b^	507 ± 1.8 ^d^	88.3 ± 0.9 ^a^
MPTVG	QYSM	2400 ± 4.5 ^d^	1596 ± 4.3 ^c^	804 ± 5.3 ^c^	3067 ± 4.8 ^d^	667 ± 1.7 ^c^	88.6 ± 0.8 ^a^
	YLY2646	2307 ± 5.3 ^e^	1661 ± 5.9 ^b^	646 ± 4.5 ^d^	3260 ± 5.3 ^ab^	953 ± 2.0 ^b^	90.2 ± 0.8 ^a^
	DG163	2559 ± 4.8 ^c^	1559 ± 5.0 ^c^	1000 ± 4.3 ^a^	3017 ± 4.7 ^e^	458 ± 2.3 ^e^	90.1 ± 1.1 ^a^
	JYZ	2240 ± 4.6 ^e^	1662 ± 4.7 ^b^	578 ± 4.3 ^e^	3295 ± 4.3 ^a^	1055 ± 2.4 ^a^	91.0 ± 1.2 ^a^

Data are expressed as mean ± SD. Values in the same column with different letters are significantly different (*p* < 0.05). PV, peak viscosity; TV, trough viscosity; BV, breakdown value; FV, final viscosity; SV, setback value; PT, pasting temperature.

## Data Availability

The data used to support the findings of this study can be made available by the corresponding author upon request.

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
