# Peer review of "Effects of Starch Molecular Structure and Physicochemical Properties on Eating Quality of Indica Rice with Similar Apparent Amylose and Protein Contents"

_foods, 2023, doi:10.3390/foods12193535_

Round 1
Reviewer 1 Report
This manuscript represents how physiochemical properties and molecular structure of starch affect eating quality of rice having similar amylose and protein content. The works presented are quite informative and the authors tried their best for the compilation of the selected content, but the manuscript needs some modifications/additions/corrections as mentioned below.
· 1. How component of starch (amylose and amylopectin) structure affects the eating quality of rice should be described in introduction section.
· 2. Elaborate the procedure mentioned in section 2.6 and 2.7.
· 3. In the manuscript “High taste value” is mentioned. It should be replaced with either good or optimum taste value.
· 4. You have selected only seven rice varieties having similar apparent amylose and protein content for the experiment and concluded that the eating quality of indica rice was affected by starch molecular structure and physicochemical properties. This kind of concluding remarks should be based on experimentation on large number of rice genotypes. Small number of genotypes cannot give confirmatory information. Additionally, contrasting rice genotypes (low AC with high protein and high AC with low protein) should be added in the experiment. Therefore, please add more genotypes with contrasting AC and protein to prove your point.
· 5. Line no. 181-182- Add the percentage value of AAC and PC in rice not beneficial for the formation of good eating quality.

The manuscript is written well but needs checking English language once as the meaning of some sentences are not clear.
Author Response
We would like to thank you very much for your careful review, comments, corrections and suggestions to our manuscript.
Based on the comments, corrections and suggestions of you, we have made a major revision. The enclosed is the point-to-point responses to comments for your review.
If you have any question, please do not hesitate to let us know.
Again, many thanks for your efforts making our manuscript much better!
Sincerely yours,
Dawei Zhu

Reviewer 2 Report
The authors have analyzed seven indica varieties for eating quality using several parameters with similar protein content and AAC. They have concluded that those varieties with low AC, low glutelin but high albumin content have better value for consumption/taste. Though eating quality is important for the 3 billion people who consume rice everyday, nutritional quality is equally, if not more, important. Since the major value of protein in rice comes because of its higher glutelin content, this portion needs to be addressed. Though protein content per se is low in rice, it has better amino acid profile with high digestive/nutritional value and is a complete food to infant and young children.
Similar studies in japonica rice are available and they need to be discussed well in the MS.
Though only a few varieties are included in the study to allow any meaningful correlation, it will be interesting to look at their correlation to understand how the biochemical profiles and sensory parameters are related with one another.
If some of the parameters are checked in any mapping population available between one of the HTVG and MLTVG varieties, the relevance of the observations and conclusions would become robust.
Author Response

(The authors gave the same response as above.)

Reviewer 3 Report
Congratulations for the excellent research work you have done on the "Effects of starch molecular structure and physicochemical propoperties on eating quality of indica rice with similar apparent amylose and protein content". This shows us that it is not only the amount of each of the main components that determine and influence the functional properties and nutritional quality of food. The size and interactions that occur between these components are also important. These results would provide new knowledge on the identification of the nutritional quality of indica rice, as well as of all those foods whose main components are starch and protein.

Author Response
We would like to thank you very much for your careful review to our manuscript.
Again, many thanks for your efforts making our manuscript much better!
Sincerely yours,
Dawei Zhu
